# Cancer-Specific miRNAs Extracted from Tissue-Exudative Extracellular Vesicles in Ovarian Clear Cell Carcinoma

**DOI:** 10.3390/ijms232415715

**Published:** 2022-12-11

**Authors:** Hiroshi Maruoka, Tomohito Tanaka, Hikaru Murakami, Hiromitsu Tsuchihashi, Akihiko Toji, Misa Nunode, Atsushi Daimon, Shunsuke Miyamoto, Ruri Nishie, Shoko Ueda, Sousuke Hashida, Shinichi Terada, Hiromi Konishi, Yuhei Kogata, Kohei Taniguchi, Kazumasa Komura, Masahide Ohmichi

**Affiliations:** 1Department of Obstetrics and Gynecology, Educational Foundation of Osaka Medical and Pharmaceutical University, 2-7 Daigakumachi, Takatsuki 569-8686, Japan; 2Translational Research Program, Educational Foundation of Osaka Medical and Pharmaceutical University, 2-7 Daigakumachi, Takatsuki 569-8686, Japan

**Keywords:** ovarian neoplasm, clear cell carcinoma, extracellular vesicles, miRNA

## Abstract

Ovarian clear cell carcinomas (OCCs) arise from endometriotic cysts that many women develop. Biomarkers for early OCC detection need to be identified. Extracellular vesicles have attracted attention as biomarker carriers. This study aims to identify cancer-specific miRNAs as novel OCC biomarkers using tissue-exudative extracellular vesicles (Te-EVs). Te-EVs were collected from four patients with OCC on one side and a normal ovary on the other side. Microarray analysis was performed to identify cancer-specific miRNAs in Te-EVs. Serum samples obtained before and after surgery from patients with OCC and atypical endometrial hyperplasia (AEH) (controls) were compared using real-time PCR to examine changes in the detected EV miRNA levels. Thirty-seven miRNAs were >2-fold upregulated on the OCC side compared with the normal ovarian side. We selected 17 miRNAs and created specific primers for 12 of these miRNAs. The levels of six EV miRNAs were significantly decreased in postoperative OCC serum compared to those in preoperative OCC serum. In contrast, no significant change was observed between the pre and postoperative values in the control group. We identified OCC tissue-specific miRNAs in the EVs secreted by OCC tissues. These EV miRNAs have potential for use as biomarkers for the early diagnosis and detection of OCC.

## 1. Introduction

According to GLOBOCAN 2020, 314,000 women are diagnosed with, and 207,000 women die from ovarian cancer every year. The incidence and mortality per 100,000 women are 6.6 and 4.2, respectively [1]. Ovarian cancer has poor prognosis because of its association with advanced cancer. Over 70% of patients are diagnosed at an advanced stage. The overall five-year survival for International Federation of Gynecology and Obstetrics (FIGO) stage III and IV ovarian cancer is only 23% [2,3]. It is difficult to detect pelvic organ diseases in patients until tumor masses become larger or are associated with a large amount of ascites, whereas early-stage ovarian cancer has few symptoms [4]. Ovarian clear cell carcinomas (OCCs) are considered to arise from endometriotic cysts that many women develop [5]. OCCs are resistant to chemotherapy and have poor prognosis [6]. In recent years, molecular-targeted therapeutic agents for homologous recombination deficiency (HRD) have been developed for high-grade serous carcinoma (HGSC) and improvements in prognosis are expected [7]. In contrast, most of the OCCs are negative for HRD. Ultrasound and serum carcinoma antigen 125 (CA125) tests have been used as ovarian cancer screening methods. Recently, the use of human epididymis protein 4 (HE4) in conjunction with CA125 has achieved high sensitivity. However, an effective screening method that leads to a decrease in ovarian cancer mortality remains to be identified [8,9]. Therefore, the development of new minimally invasive and reliable methods, such as the detection of circulating biomarkers, is essential for the early detection of cancer [10]. In recent years, liquid biopsy has been used to measure biomarkers and diagnose diseases. It is a minimally invasive method that uses body fluids instead of tissue. Using liquid biopsy, circulating tumor cells (CTCs) and cancer cell-derived DNA (ctDNA) have been analyzed [11,12]. However, since CTCs and ctDNA are detected when carcinoma cells metastasize, they may not be suitable for early diagnosis [11,13]. Extracellular vesicles (EVs) have the potential to overcome these limitations. They are small vesicles with diameters of 30–150 nm that are released from various cell types, including cancer cells. They can be obtained from several biological fluids, including urine, plasma, and serum [14]. EVs contain miRNA, mRNA, long noncoding RNA, and proteins. They affect many signaling pathways after transport to recipient cells [15]. Recently, it was reported that EVs play an important role in mediating intercellular communication in various aspects of tumorigenesis, including angiogenesis, immunosuppression, and metastasis; hence, they can be considered potential circulating biomarkers for tumor diagnosis and prognosis [16]. MiRNAs are 19 to 25 nucleotide-long endogenous molecules which are integral to regulating gene expression. Aberrant miRNA expression profiles have been described in several cancers and have been implicated as useful biomarkers which may aid cancer diagnostic and treatment [17]. In the current study, EVs obtained from OCC and normal ovaries in the same patients were compared to identify cancer-specific miRNAs in EVs released from OCC cells. Furthermore, serum samples obtained before and after surgery from patients with OCC and atypical endometrial hyperplasia (AEH) (controls) were compared to detect cancer-specific miRNAs in EVs. Cancer-specific miRNAs in EVs may be useful for the early detection of OCCs.

## 2. Results

### 2.1. Isolation of Extracellular Vesicles from Cultured Patient-Derived Tissue

Western blot analysis and electron microscopy were performed to confirm that EVs were obtained in our procedures. Electron microscopy was performed to Te-EVs obtained from the cancer specimens in case 4. Western blot analysis was performed to all Te-EVs obtained from cancer and normal specimens in each case (Appendix A). However, we did not perform them on serum samples because the amount of sample was small and limited. The protein concentrations of isolated Te-EVs and serum EVs, and the RNA concentrations extracted from isolated Te-EVs and serum EVs are shown in Appendix A. Western blot analysis showed the EV-specific protein expression of CD9, 63, and 81 (Figure 1A). Electron microscopy showed the presence of EVs in the culture medium, which were separated using ultracentrifugation (Figure 1B).

### 2.2. MiRNA Microarray Analysis of Te-EVs

MiRNA microarray analysis was performed to Te-EVs obtained from OCC specimens and normal ovaries. Figure 2 shows the expression of 4603 miRNAs in *Homo sapiens*. In each case, the levels of several miRNAs were two-fold higher in the EVs released from cancer cells compared with those in normal ovaries: An amount of 255 miRNAs in case 1 (Figure 2A), 342 miRNAs in case 2 (Figure 2B), 90 miRNAs in case 3 (Figure 2C), and 272 miRNAs in case 4 (Figure 2D) were measured. Among them, 37 miRNAs were detected in all four cases (Figure 2E). Figure 3 shows the heat map analysis of miRNA expression in OCC and normal regions. In the hierarchical clustering analysis, miRNAs with high expression were color-coded in red. Those with low expression were color-coded in blue. Thirty-seven miRNAs were clustered into high- and low-expression groups. Appendix A shows the detailed information on the 37 miRNAs. We selected 17 of these 37 miRNAs (miR−200c−3p, miR−200a−5p, miR−200b−3p, miR−182−5p, miR−203a, miR−200b−5p, miR−183−5p, miR−141−3p, miR−1301−3p, miR−421 miR−30a−5p, miR−30a−3p, miR−21−5p, miR−210−3p, miR−652−3p, miR−330−3p), which were overexpressed in OCCs compared with normal ovaries by eight-fold or higher.

### 2.3. Comparison of Serum miRNAs in EVs before and after Surgery

One serum sample was obtained before and after surgery. We calculated the changes in serum miRNAs in EVs before and after surgery in patients with OCC. Six samples were obtained from patients with OCC (Table 1) and four samples were obtained from patients with atypical endometrial hyperplasia (AEH) as a control (Table 2). Of the 17 selected miRNAs, specific primers were created for 12 miRNAs. Figure 4A shows the changes in miRNAs in serum EVs before and after surgery in patients with OCC. Figure 4B shows the changes in miRNAs in serum EVs before and after surgery in patients with AEH. In patients with OCC, six miRNAs (miR−200a−3p, miR−200b−3p, miR−200c−3p, miR200a−5p, miR−200b−5p, and miR−30a−5p) were significantly downregulated after surgery. In contrast, there were no significant differences in the 12 miRNAs before and after surgery in patients with AEH.

## 3. Discussion

In this study, we identified six OCC tissue-specific miRNAs (miR−200a−3p, miR−200b−3p, miR−200c−3p, miR200a−5p, miR−200b−5p, and miR−30a−5p) in EVs secreted from OCC. These EV-derived miRNAs may serve as potential biomarkers for the early diagnosis and detection of ovarian cancer.

For the past two decades, miRNAs have been investigated for their roles in many cancers. They modulate gene expression and play significant roles in biological pathways, influencing proliferation, apoptosis, cell cycle, invasion, and migration. Interest is focused on their use as biomarkers for the early detection, diagnosis, and prognosis of cancers. They are released by cancer cells and detectable in all human biofluids, including plasma, serum, and urine. Many cancer-specific circulating miRNAs have been reported and are expected to serve as viable biomarkers for the early detection of various cancers [17,18,19,20,21].

Several authors have reported miRNAs in EVs extracted from the serum or plasma of patients with ovarian cancer [22,23,24]. Taylor et al. analyzed miRNAs in Te-EVs and serum EVs extracted from patients with high-grade serous ovarian carcinoma and benign adenomas. They concluded that eight miRNAs (miR−141, miR−21, miR−200a, miR−200b, miR−200c, miR−214, miR−205, and miR−203) were highly expressed in Te-EVs and serum EVs of patients with ovarian cancer [22]. Meng et al. compared miRNAs in serum EVs between patients with OC and healthy women. They found that the expression of four miRNAs (miR−373, miR−200a, miR−200b, and miR−200c) in patients with ovarian cancer was higher than that in healthy women [23]. Furthermore, Pan et al. reported miRNAs in plasma-derived EVs of ovarian cancer patients, ovarian cyst patients, and healthy women using quantitative TaqMan real-time PCR and miRNA array cards. The results showed that four miRNAs (miR−21, miR−100, miR−200b, and miR−320) were highly expressed in the EVs of patients with ovarian cancer compared to the EVs of healthy women [24]. In the current study, we detected cancer-specific miRNAs in EVs by comparing cancerous tissues and normal ovaries in the same patient.

Several studies have reported the presence of cancer-specific miRNAs in OCC. In these studies, miRNAs were extracted directly from cell lines and tissues, or from EVs secreted by cell lines [25,26,27]. Wyman et al. compared miRNAs extracted from ovarian serous, endometrial, and clear cell carcinoma tissues with those extracted from normal ovarian surface epithelial cells. They found that the expression of six miRNAs (miR−142−3p, miR−195, miR−200a, miR−200b, miR−200c, and miR−338−3p) was higher in the three cancer types than those in normal ovarian surface epithelial cells. In addition, they showed that four miRNAs (miR−486−5p, miR−144, miR−30a, and miR−199a−5p) are specifically expressed in OCC [25]. Yanaihara et al. reported the miRNA expression profile of cryostat sections of surgically removed HGSC and OCC tissues. They found that five miRNAs (miR−132, miR−9, miR−126, miR−34a, and miR−21) were overexpressed in OCC [26]. Horie et al. extracted and profiled miRNAs from EVs from four histological types of ovarian cancer cell lines (serous, endometrial, mucinous, and clear cells). They reported that three miRNAs (miR−21−5p, miR29a−3p and miR−30d−5p) were highly expressed in OCC [27]. In the current study, the expression levels of miR−200c−3p, miR−200a−3p, miR200b−3p, miR−200b−5p, miR−200a−5p, miR−30a−5p, miR−30b−5p, and miR−21−5p in Te-EVs extracted from OCC tissues were significantly higher than those in normal ovaries. The expression of miR30a−5p, miR−200a−5p, miR200a−3p, miR−200b−5p, miR−200b−3p, and miR−200c−3p in serum EVs decreased after surgery. However, that of miR−21−5p did not. Although miR−21 is reported to be a biomarker for lung cancer, it has also been reported to be a marker for many non-neoplastic diseases, such as myocardial infarction, hepatitis, pneumonia, and Crohn’s disease [28,29]. These eight miRNAs have not been previously reported as potential biomarker for OCC.

However, the functions of these miRNAs are not yet fully understood. Among the 37 miRNAs detected in this study, some are considered to belong to the miR−200 family, including miR−200a, miR−200b, miR−200c, and miR−141. The miR−200 family comprises five members: miR−200a, miR−200b, miR−200c, miR−141, and miR−429 [30]. The miR−200 family is known to play a role in regulating epithelial–mesenchymal transition (EMT) in a variety of cancers, including breast, pancreatic, colorectal, prostate, and ovarian cancer [31,32,33,34]. In the current study, the levels of miR−200a−5p, miR−200a−3p, miR−200b−5p, miR−200b−3p, miR−200c−3p, and miR−141−3p were higher in Te-EVs extracted from cancer tissue compared to those extracted from normal ovarian tissue in identical patients. In previous reports, the miR−200 family was highly expressed in ovarian cancer as a whole. Furthermore, the authors concluded that the miR−200 family plays a significant role in ovarian cancer. Patients with miR−200c overexpression tend to have early-stage diseases and favorable prognoses [35].

Some authors have reported that miR−30a−5p functions as an anti-oncogene in several malignancies [36,37,38]. As previously reported, miR−30a−5p inhibits migration, invasion, and EMT processes by downregulating ITGB3 in colorectal cancer [36]. MiR−30a−5p inhibition promotes prostate cancer cell growth by regulating PCLAF [37]. Inhibition of miR−30a−5p expression in a gallbladder cancer cell line increased cell proliferation, migration, and invasiveness [38]. Nude mice transplanted with a cell line that strongly expressed miR−30a−5p had a reduced metastatic rate. Ye et al. reported that E2F transcription factor 7 (E2F7) is a target of miR−30a−5p [38]. In most cancers, miR−30a−5p regulates proliferation, invasion, and EMT as a suppressor. However, there are several reports on the effects of miR−30a−5p on ovarian cancer. Liu et al. extracted and examined miRNAs from cisplatin-resistant and -sensitive ovarian cancer cell lines. The results showed that miR−30a−5p was more highly expressed in cisplatin-resistant cell lines than in cisplatin-sensitive cell lines. They also found that high expression of miR−30a−5p promoted cell proliferation and colony formation and promoted cell migration and invasion [39]. However, the function of miR−30a−5p in OCC remains unclear.

This study had several limitations. First, the control blood samples were obtained from AEH patients; AEH may have an influence on miRNAs. Second, the blood samples and OCC tissue specimens analyzed were obtained from two different groups of patients due to a low RNA quality in blood samples obtained from the patients who provided tissue specimens. Third, the sample size was too small to prove that miRNA in EVs is a useful biomarker. It is not clear whether other differences between patients (age, infection, other tumor, etc.) would not result in changes in these miRNAs. Therefore, an analysis of a larger sample of OCC patients and age matched individuals without tumors is needed.

## 4. Materials and Methods

### 4.1. Patients and Tissue Samples

Specimens were obtained from four patients with OCC who underwent surgery between 2018 and 2019 at Osaka Medical and Pharmaceutical University. All four patients had an OCC on one side and a normal ovary on the opposite side. The surgically removed specimens were washed thoroughly with 500 mL of saline solution. Then, tissue fragments (1 cm squares; 2 mm thick) were excised including the OCC and the normal ovary. The obtained fragments were individually incubated. Table 3 shows the characteristics of the four patients with OCC. The patients were aged 42–67 years and had an OCC on one side and a normal ovary on the opposite side. The disease and its diagnosis were confirmed pathologically. Three patients had stage IC disease and one patient had stage IIIB disease. Intraperitoneal cytology results were negative in all patients. The follow-up periods were 26 to 45 months. One patient exhibited recurrence 16 months after surgery. Blood samples were obtained from patients with OCC and AEH who underwent surgery between 2019 and 2020 at Osaka Medical and Pharmaceutical University. All patients with OCC underwent total abdominal hysterectomy, bilateral salpingo-oophorectomy, pelvic and para-aortic lymphadenectomy, and omentectomy. The disease was completely dissected in all patients. Patients with AEH, who were considered as controls, underwent total laparoscopic hysterectomy with bilateral salpingo-oophorectomy. Blood samples were obtained from the patients one week before and one month after surgery. Blood samples were collected in the morning, at least 10 h after the last meal. Water intake was allowed during the fasting period. Blood was collected in plain 7 mL serum tubes (VP-P070K, Terumo, no additives). Tumors were staged according to the International Federation of Gynecology and Obstetrics (FIGO) staging system 2014. All patients were Asian in race. The study protocol was approved by the ethical guidelines of the 1975 Declaration of Helsinki, as revised in 1983, the Osaka Medical and Pharmaceutical University Clinical Research Review Board (IRB protocol number: 2191) and the participants provided written informed consent. Inclusion criteria were: lack of preoperative chemotherapy, hormonal therapy, or radiation therapy, and no other inflammatory diseases. Exclusion criteria were: malignancy other than ovarian cancer, chronic or acute inflammatory disease, infectious disease, autoimmune disease, cardiovascular disease, and pregnancy or lactation in the past 6 months. No anticoagulants were used and all samples were stored at −80 °C until processing. Repeated thawing during storage was avoided to ensure sample quality.

### 4.2. Purification of Extracellular Vesicles

Figure 5 shows the flow chart of incubation of ovarian tissue, filtering, collection of EVs using ultracentrifugation, miRNA extraction, and microarray analysis. Following excision, tissue samples were immediately immersed in 9 mL serum-free medium (FUJIFILM Wako Pure Chemical Corporation, Osaka, Japan) with 1 mL exosome-depleted FBS (System Biosciences, Palo Alto, CA, USA), collected in 100-mm dishes individually, and stored for 3 h at 37 °C in a humidified incubator containing 5% CO_2_. The tissue-immersed medium was filtered using a 0.22 μm filter (Merck Millipore, Bedford, MA, USA) to remove cell debris and centrifuged at 2000× *g* for 30 min, 4 °C to remove cells and dead cells, and centrifuged 16,000× *g* for 30 min, 4 °C to remove cell debris. Blood samples were centrifuged at 2500× *g* for 10 min with 7 mL of blood within 1 h of collection. This centrifugation was performed to remove as many impurities as possible, while using a centrifugal speed and centrifugal time that does not destroy the blood cells. The supernatant was centrifuged at 20,000× *g* for 5 min. This centrifugation removed the leukocytes. Processes were completed within 2 h of blood collection to prevent elution of blood components into plasma and serum and stored at −80 °C until the next process. To pellet EVs from the collected supernatant, ultracentrifugation was performed using an Optima XE-100 (Beckman Coulter, Brea, CA, USA), SW41 T1 (Beckman Coulter), and Ultra-Clear tube (Beckman Coulter) at 100,000× *g* for 70 min, 4 °C. EVs were collected in phosphate-buffered saline to make a total of 150 μL, placed in a 1.5 mL collection tube, and stored at −80 °C. The protein concentration content of EVs was measured using the Pierce^TM^ BCA protein assay kit (Thermo Fisher Scientific, Waltham, CA, USA). Western blotting and electron microscopy confirmed the EV extraction.

### 4.3. Western Blot Analysis

EVs samples were dissolved in radioimmunoprecipitation assay buffer (RIPA buffer; 25 mM Tris-HCl pH 7.6, 150 mM NaCl, 1% deoxycholate sodium, 1% NP-40, and 0.1% sodium dodecyl sulfate, Thermo Fisher Scientific). An amount of 5.0 µg of EV samples was placed in each lane. The lysates were separated using 5–20% sodium dodecyl sulfate–polyacrylamide gel electrophoresis and transferred to polyvinylidene difluoride membranes. The membranes were blocked in 10% bovine serum albumin in 1X Tris-buffered saline and incubated overnight with specific primary antibodies against β-actin (clone 13E5, 1:1000 dilution, Cell Signaling Technology, Boston, MA, USA), CD9 (clone ab223052, 1:1000 dilution; Abcam, Cambridge, MA, USA), CD63 (clone sc-5275, 1:200 dilution; Santa Cruz Biotechnology, Dallas, TX, USA), and CD81 (clone sc-166029, 1:750 dilution; Santa Cruz) at 4 °C. Then, the membranes were incubated with horse radish peroxidase-conjugated secondary antibodies for 1 h. Bands were visualized using an enhanced chemiluminescence agent (ECL Plus; GE Healthcare Life Sciences, Pittsburgh, PA, USA).

### 4.4. Transmission Electron Microscopy

EVs (1 μg) were incubated with poly-l-lysine solution-coated beads (φ 3.10 μm; Merck Millipore). After drying, the beads were washed and fixed in 1.25% glutaraldehyde in 0.1 M phosphate buffer (PB; pH 7.4). Then, they were washed with PB and fixed in 1% osmium tetroxide for 40 min. After washing with PB, the beads were gradually dehydrated using a graded series of ethanol washes. The platinum–palladium was evaporated and vapor deposition was performed. Next, scanning electron microscopy analysis (SEM; S-5000; HITACHI, Tokyo, Japan) was performed.

### 4.5. Microarray Analysis of miRNA

Initially, two total RNA extractions from EVs were performed using the miRNeasy Mini Kit according to the manufacturer’s protocol (Qiagen GmbH, Hilden, Germany). The concentration and purity of RNA were assessed using spectrophotometry (NanoDrop ND-1000; Thermo Fisher Scientific, Wilmington, DE, USA), and RNA integrity was checked using a 2100 Bioanalyzer (Agilent Genomics, Santa Clara, CA, USA) before microarray hybridization. The samples were hybridized, washed, and stained using a GeneChip Hybridization Oven 640, GeneChip Fluidics Station 450 (Affymetrix; Thermo Fisher Scientific Inc., Waltham, MA, USA), and Affymetrix GeneChip^®^ miRNA 4.0 (Affymetrix; Thermo Fisher Scientific Inc.) using the protocol provided by Affymetrix. The GeneChip miRNA 4.0 array contains over 30,000 probes from Sanger miRBase v.20. The array was analyzed using the Transcriptome Viewer software (KURABO, Tokyo, Japan). The miRNA signal values were standardized by global normalization (log transformation of data and median alignment). Relative expression levels for each miRNA were validated by one-way analysis of variance or *t*-test (*p* < 0.05). MicroRNAs whose expression levels showed at least a 2-fold difference (|log2-fold change| > 1 and *p* < 0.05) between the test and control samples were analyzed.

### 4.6. Extraction of EV miRNAs

Total miRNAs were extracted from the suspended EVs using the miRVana™ miRNA Isolation Kit according to the manufacturer’s recommendations (#AM1560; Life Technologies, Carlsbad, CA, USA).

### 4.7. Quantitative Reverse Transcription-Polymerase Chain Reaction (qRT-PCR) of miRNAs from Serum EVs before and after Surgery

qRT-PCR was performed using a StepOnePlus Real-Time PCR System (Applied Biosystems, Foster City, CA, USA). Pre and postoperative OCC and AEH blood samples were used. The RNA concentration was adjusted to 2.0 ng/μL. The miRNAs were reverse transcribed using a microRNA reverse transcription kit (Thermo Fisher Scientific Inc.) and qPCR was performed according to the manufacturer’s instructions (Thermo Fisher Scientific Inc.). The following primers were used for PCR: miR−21−5p, miR−30a−5p, miR−30c−5p, miR−141−3p, miR−182−5p, miR−183−5p, miR−200a−5p, miR−200a−3p, miR−200b−5p, miR−200b−3p, miR−200c−3p, miR−210−3p, miR−652−3p, and RNU48 (#4427975; Thermo Fisher Scientific Inc.). RNU48 was used as the normalization control. qPCR of miRNA was performed three times in triplicate wells. The relative expression of miRNAs was calculated using the 2^−ΔΔCq^ method.

### 4.8. Statistical Analyses

Statistical analyses were performed using JMP Pro version 14.2.0 (SAS Institute Japan, Tokyo, Japan). The Wilcoxon test was used to compare continuous variables. Statistical significance was set at *p* < 0.05.

## 5. Conclusions

We identified the following OCC tissue-specific miRNAs in EVs secreted by tissues: miR−200a−3p, miR−200b−3p, miR−200c−3p, miR−200a−5p, miR−200b−5p, and miR−30a−5p. These miRNAs may serve as potential OCC-specific biomarkers.

## Figures and Tables

**Figure 1 ijms-23-15715-f001:**
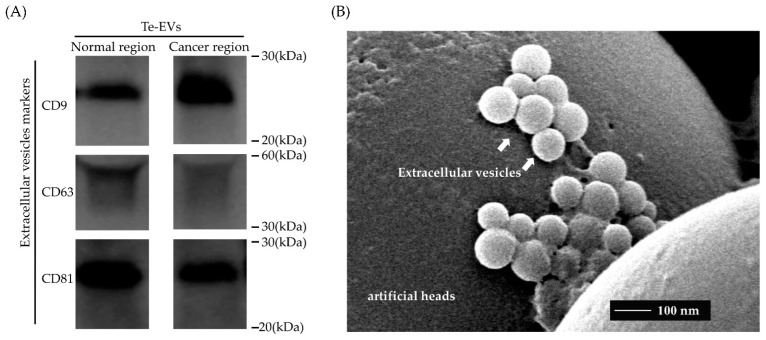
Confirmatory analysis of tissue-exudative extracellular vesicles (Te-EVs). EVs were isolated from serum-free cultures of tissue removed from the cancerous and normal ovaries of patients with ovarian clear cell carcinomas. (**A**) Western blot analysis showing EV-specific protein expression of CD9, CD 63, and CD81 in case 4. (**B**) A representative image of Te-EVs obtained from cancerous specimens in case 4 using transmission electron microscopy showing that the isolated clusters were mainly circular or oval membrane vesicles with a size of 30–100 nm, uniform in appearance, and exhibited the characteristic appearance of EVs.

**Figure 2 ijms-23-15715-f002:**
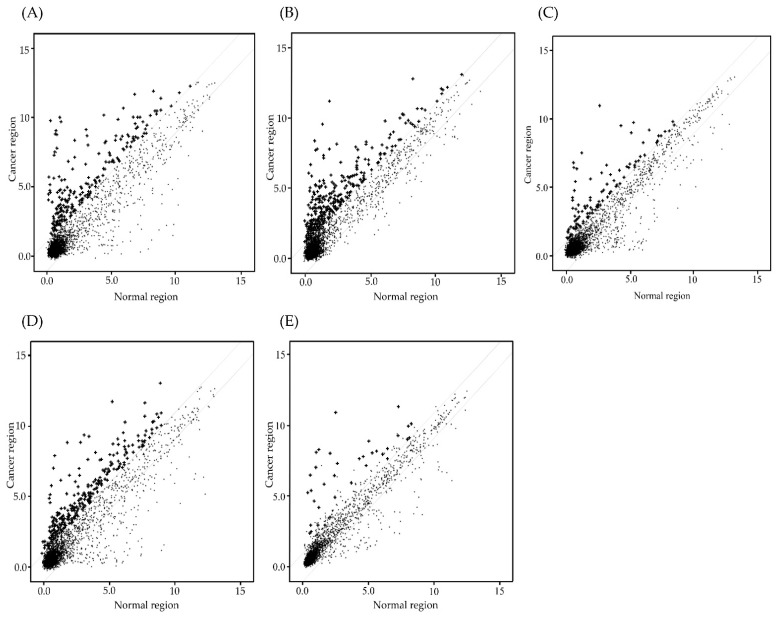
MiRNA microarray analysis was performed to Te-EVs obtained from OCC specimens and normal ovaries. Scatter plot of miRNA signal values showing the miRNA expression variation between the cancerous and normal samples. The values spotted in the x- and y-axes represent the normalized signals of samples in the two groups (log2-scaled). The solid lines represent fold changes. The miRNAs above the upper solid line and below the lower solid line are those with a >2-fold change in expression levels between the cancerous and normal-appearing tissues. The cross marks indicate that the miRNAs are two-fold or higher overexpressed in the cancerous region compared with the normal region. (**A**) 255 miRNAs were detected in case 1. (**B**) 342 miRNAs were detected in case 2. (**C**) 90 miRNAs were detected in case 3. (**D**) 272 miRNAs were detected in case 4. (**E**) 37 miRNAs were detected in all four cases.

**Figure 3 ijms-23-15715-f003:**
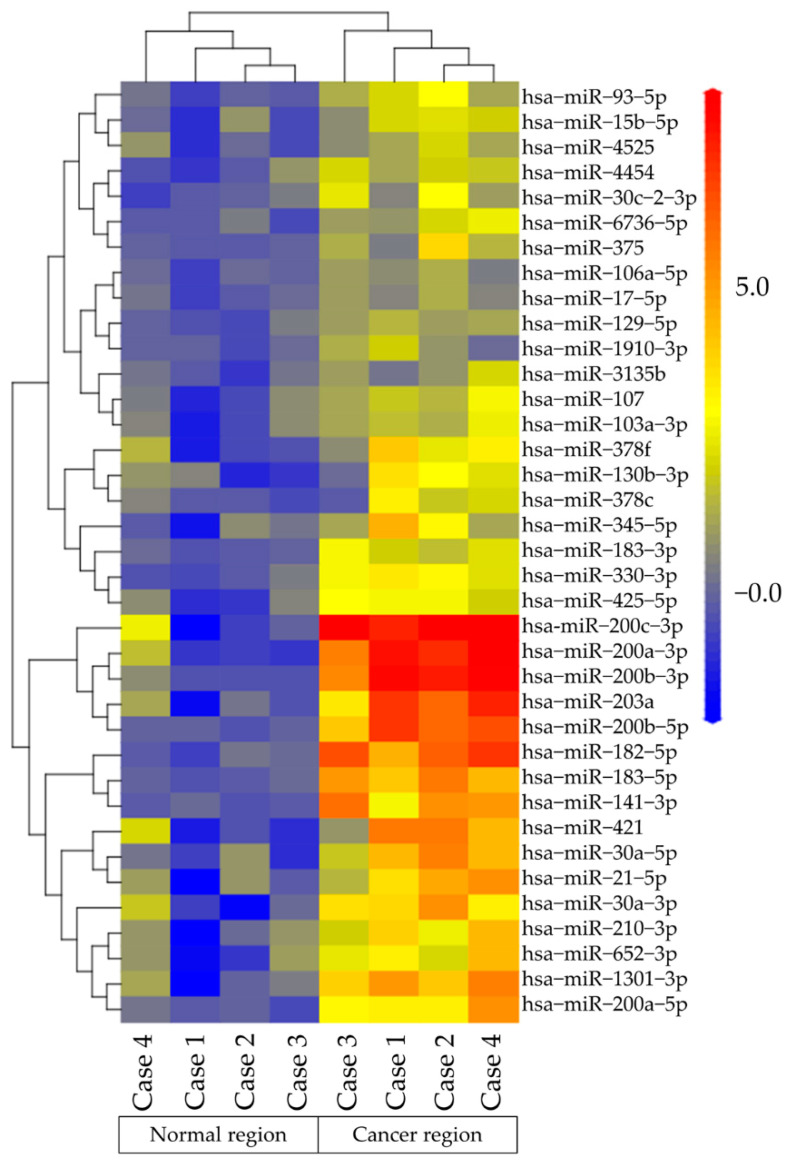
Heat map of the expression profiles of the 37 miRNAs based on microarray analysis in Te-EVs obtained from cancerous and normal tissues. The color scale is blue (low intensity), yellow (medium intensity), and red (high intensity).

**Figure 4 ijms-23-15715-f004:**
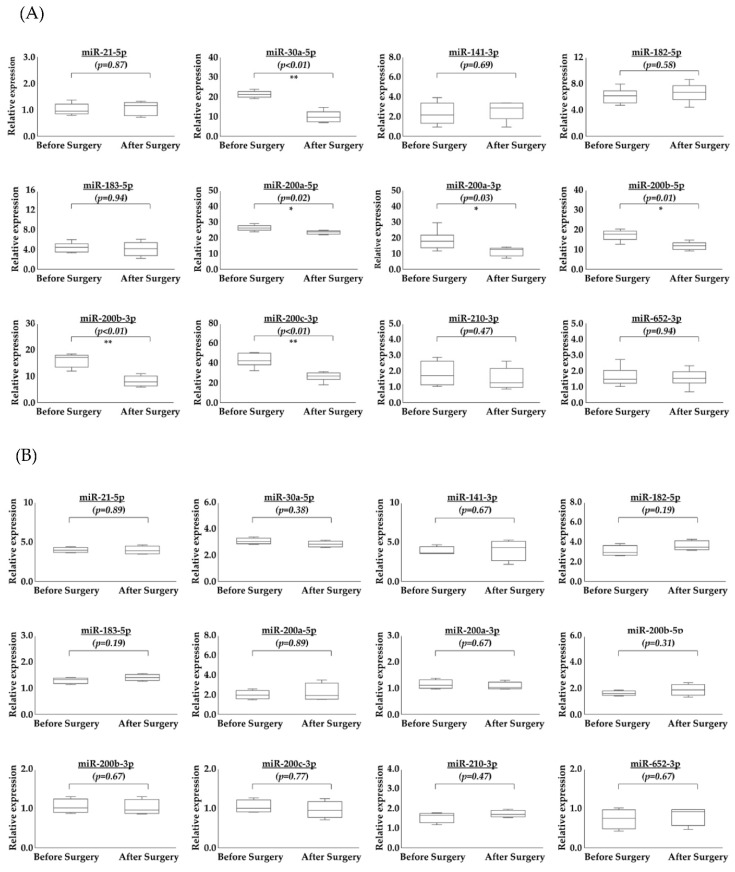
miRNA levels in serum extracellular vesicles (EVs) of patients with ovarian clear cell carcinoma. One serum sample was obtained before, and one after, surgery. (**A**) Six miRNAs (miR−30a−5p, miR−200a−5p, miR−200a−3p, miR−200b−5p, miR−200b−3p, miR−200c−3p) were significantly (** *p* < 0.01, * *p* = 0.0202, * *p* = 0.0306, * *p* = 0.0131, ** *p* < 0.01, ** *p* < 0.01 respectively) downregulated after surgery. (**B**) No significant differences were observed between samples obtained before and after surgery in patients with atypical endometrial hyperplasia patient (controls).

**Figure 5 ijms-23-15715-f005:**
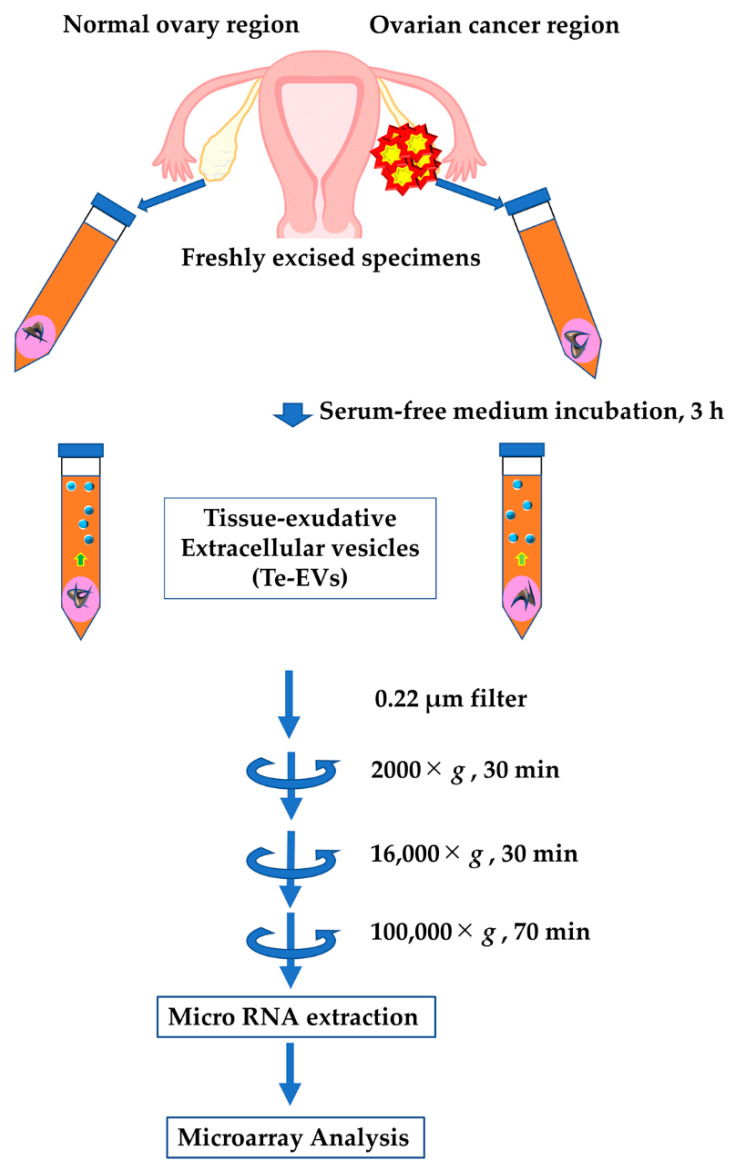
Isolation of Te-EVs and microarray search for miRNAs.

**Table 1 ijms-23-15715-t001:** Characteristics of six patients with ovarian clear cell carcinomas.

Case	Age (Years)	BMI (kg/m^2^)	FIGO Stage	Intraperitoneal Cytology	Follow-Up (Months)	Recurrence	Menopause	Other Disease	Medication
1	54	20.1	IA	Negative	29	No	Yes	No	No
2	54	20.2	IA	Negative	26	No	Yes	No	No
3	64	21.4	IC1	Negative	29	No	Yes	No	No
4	65	26.8	IIIA	Positive	27	Yes	Yes	No	No
5	66	22.3	IIIB	Positive	33	Yes	Yes	No	No
6	70	20.1	IA	Negative	32	No	Yes	HT	Ca blocker

BMI, Body Mass Index; FIGO, the International Federation of Gynecology and Obstetrics; HT, Hypertension.

**Table 2 ijms-23-15715-t002:** Characteristics of four patients with atypical endometrial hyperplasia.

Case	Age (Years)	BMI (kg/m^2^)	Intraperitoneal Cytology	Follow-Up (Months)	Recurrence	Menopause	Other Disease	Medication
1	54	24.4	Negative	26	No	Yes	No	No
2	57	25.3	Negative	39	No	Yes	No	No
3	57	20.2	Negative	25	No	Yes	No	No
4	58	26.1	Negative	25	No	Yes	No	No

BMI, Body Mass Index.

**Table 3 ijms-23-15715-t003:** Characteristics of four patients with ovarian clear cell carcinomas.

Case	Age (Years)	BMI (kg/m^2^)	FIGO Stage	Tumor Origin	Intraperitoneal Cytology	Follow-Up (Months)	Recurrence	Menopause	Other Disease	Medication
1	42	17.9	IC1	Right ovary	Negative	45	No	No	No	No
2	67	17.5	IC1	Right ovary	Negative	43	No	Yes	HT	Ca blocker
3	55	21.8	IIIB	Left ovary	Negative	26	No	Yes	No	No
4	45	16.7	IC2	Right ovary	Negative	40	Yes	No	No	No

BMI, Body Mass Index; FIGO, the International Federation of Gynecology and Obstetrics; HT, Hypertension.

## Data Availability

The data that support the findings of this study are available upon request from the corresponding author. The data are not publicly available because of privacy restrictions.

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
