# Peer review of "Cancer-Specific miRNAs Extracted from Tissue-Exudative Extracellular Vesicles in Ovarian Clear Cell Carcinoma"

_ijms, 2022, doi:10.3390/ijms232415715_

Round 1
Reviewer 1 Report
This manuscript describes identification of miRNAs that may be used as biomarkers for OCC in patients. The authors showed that some of these miRNAs are present in EVs at a higher level before surgical removal of the tumor than after removal. From this they conclude that the miRNAs can be used as biomarkers for the presence of OCC.
There are some conceptual problems with this line of thinking. They showed that surgical removal, on average, resulted in lower levels of the miRNAs. However, they did not show that it can be used as a biomarker. Although it is not completely clear how the qRT-PCRs were done (what was used for normalization, how was it exactly quantified), the maximum they can show with this approach is that in the same patient the level decreases after removal of the tumor. However, I did not find any evidence that on the level of the individual patient there is indeed such a difference that it can be used to identify the presence of a tumor. Furthermore, it is not clear whether other differences between patients (age, infection, other tumor, etc.) would not also result in changes in these miRNAs. In the absence of this, it is a very preliminary analysis without any evidence that they can be used as biomarker. Therefore, I would at least expect analysis of a cohort of OCC patients and age-matched individuals without a tumor. The authors should then present these data not as average over the group, but on a per patient level to get some evidence that it can indeed be used to identify patients from a population. Definitely, the claim that it can be used for early detection is not supported by the data.
Furthermore, the authors should provide more experimental detail. For example, it is not clear how microarrays were analyzed. The authors just mention 2-fold higher expression in OCC than normal tissue, but do not provide any information on the false discovery rate they used as threshold. Also details on qRT-PCR are lacking.
Therefore, I would advise the authors to extend their analysis to really answer the question which miRNA could be used as biomarker, rather than just publishing a very preliminary analysis.
Author Response
We appreciate the time and effort provided by the editor and referees in reviewing our manuscript. We have addressed all issues indicated in the review report and hope that the revised version meets the journal's requirements for publication.
Response to Comments from Reviewer 1:
Comment 1: There are some conceptual problems with this line of thinking. They showed that surgical removal, on average, resulted in lower levels of the miRNAs. However, they did not show that it can be used as a biomarker. Although it is not completely clear how the qRT-PCRs were done (what was used for normalization, how was it exactly quantified), the maximum they can show with this approach is that in the same patient the level decreases after removal of the tumor. However, I did not find any evidence that on the level of the individual patient there is indeed such a difference that it can be used to identify the presence of a tumor. Furthermore, it is not clear whether other differences between patients (age, infection, other tumor, etc.) would not also result in changes in these miRNAs. In the absence of this, it is a very preliminary analysis without any evidence that they can be used as biomarker. Therefore, I would at least expect analysis of a cohort of OCC patients and age-matched individuals without a tumor. The authors should then present these data not as average over the group, but on a per patient level to get some evidence that it can indeed be used to identify patients from a population. Definitely, the claim that it can be used for early detection is not supported by the data.
Response: As you suggested, we do not show that miRNA in EVs can be used as a biomarker for early detection. Those processes are time- and cost-intensive. Unfortunately, we could not perform an analysis of a cohort of OCC patients and age-matched individuals without a tumor. However, we intend to perform the same in our subsequent study. Based on your suggestions, we revised the title and have added the following sentences to the Discussion in a new paragraph on limitations: “Third, the sample size was too small to prove that miRNA in EVs is a useful biomarker; it is not clear whether other differences between patients (age, infection, other tumor, etc.) would not also result in changes in these miRNAs. Therefore, an analysis of a larger sample of OCC patients and age matched individuals without tumors is needed.” (page 9, line 239)
We also added these details about qRT-PCR to the Materials and methods: “Pre- and post-operative OCC and AEH blood samples were used. The miRNAs were reverse transcribed using a microRNA reverse transcription kit (Thermo Fisher Scientific Inc.) and qPCR was performed according to the manufacturer's instructions (Thermo Fisher Scientific Inc.). MiR-16 was used as the normalization control. qPCR of miRNA was performed three times in triplicate wells. The relative expression of miRNAs was calculated using the 2-ΔΔCq method.” (page 12, line 335)
Comment 2: Furthermore, the authors should provide more experimental detail. For example, it is not clear how microarrays were analyzed. The authors just mention 2-fold higher expression in OCC than normal tissue, but do not provide any information on the false discovery rate they used as threshold. Also details on qRT-PCR are lacking.
Response: According to your suggestions, we added the following sentences about microarray analysis and the above sentences about qRT-PCR: “The miRNA signal values were standardized by global normalization (log transformation of data and median alignment). Relative expression levels for each miRNA were validated by one-way analysis of variance or t-test (p<0.05). MicroRNAs whose expression levels showed at least a 2-fold difference (|log2-fold change|>1 and p<0.05) between the test and control samples were analyzed.” (page 12, line 319)
Comment 3: Therefore, I would advise the authors to extend their analysis to really answer the question which miRNA could be used as biomarker, rather than just publishing a very preliminary analysis.
Response: According to your suggestions, we added the above sentences about miRNA as a biomarker in the discussion section. (page 8, line 162)

Reviewer 2 Report
I really liked your study. Beautifully done, well followed and very interesting conclusions.
1. The authors aim to identify through this study new circulating biomarkers that can be used to identify the early stages of cancer, particularly in Ovarian clear cell carcinomas (OCCs). Therefore, the authors raise the question whether cancer-specific miRNAs in EVs released from OCC cells can be such a biomarker.2. Medical articles and profile studies that address this problem are practically missing. Those that exist in the literature address the overall problem and do not target OCCs 3. The study strictly addresses the identification as a biomarker of cancer-specific miRNAs in EVs in OCCs, without identifying other studies with this profile 4. The methodology used can add additional criteria differences between patients (number with statistical relevance, age, infection, other tumours synchronously or in history, etc.). I draw your attention, it is a preliminary study, with specific limitations, which initiates an area of great interest and which can serve as a basis for further studies. 5. Being an initial study, beginning in the problem, having also specific limitations such number of patients, working method, etc. it can only lead to limited conclusions. Practically, the study draws attention to the use of cancer-specific miRNAs in EVs released from OCC cells. In other words, it opens the door, the opportunity, for much larger studies to prove the value of this biomarker
6.The bibliography has reached its limits, the references on this topic being extremely few
7. The bibliography has reached its limits, the references on this topic being extremely few
Author Response
We appreciate the time and effort provided by the editor and referees in reviewing our manuscript. We have addressed all issues indicated in the review report and hope that the revised version meets the journal's requirements for publication.
Response to Comments from Reviewer 2:
Comment 1: I really liked your study. Beautifully done, well followed and very interesting conclusions.
- The authors aim to identify through this study new circulating biomarkers that can be used to identify the early stages of cancer, particularly in Ovarian clear cell carcinomas (OCCs). Therefore, the authors raise the question whether cancer-specific miRNAs in EVs released from OCC cells can be such a biomarker.
2. Medical articles and profile studies that address this problem are practically missing. Those that exist in the literature address the overall problem and do not target OCCs 3. The study strictly addresses the identification as a biomarker of cancer-specific miRNAs in EVs in OCCs, without identifying other studies with this profile 4. The methodology used can add additional criteria differences between patients (number with statistical relevance, age, infection, other tumours synchronously or in history, etc.). I draw your attention, it is a preliminary study, with specific limitations, which initiates an area of great interest and which can serve as a basis for further studies. 5. Being an initial study, beginning in the problem, having also specific limitations such number of patients, working method, etc. it can only lead to limited conclusions. Practically, the study draws attention to the use of cancer-specific miRNAs in EVs released from OCC cells. In other words, it opens the door, the opportunity, for much larger studies to prove the value of this biomarker
6.The bibliography has reached its limits, the references on this topic being extremely few
Response: Thank you for your kind advice on our manuscript. According to your suggestions, we cited an additional study regarding ovarian cancer biomarkers in the Introduction and added the following discussion of miRNAs as cancer biomarkers to the Discussion: “For the past two decades, miRNAs have been investigated for their roles in many cancers. They modulate gene expression and play significant roles in biological pathways, influencing proliferation, apoptosis, cell cycle, invasion, and migration. Interest is focused on their use as biomarkers for the early detection, diagnosis, and prognosis of cancers; they are released by cancer cells and detectable in all human bio-fluids, including plasma, serum, and urine. Many cancer-specific circulating miRNAs have been reported, and are expected to serve as viable biomarkers for the early detection of various cancers [17-21].” (page 8, line 162)
We also added this to a new paragraph on limitations: “ Third, the sample size was too small to prove that miRNA in EVs is a useful biomarker; it is not clear whether other differences between patients (age, infection, other tumor, etc.) would not result in changes in these miRNAs. Therefore, an analysis of a larger sample of OCC patients and age matched individuals without tumors is needed.” (page 9, line 239)

Reviewer 3 Report
The article identifies potential ovarian cancer-specific biomarkers for early diagnosis.
Numerous studies have evaluated expression profile microRNAs (miRNAs) in tissue and serum samples of ovarian cancer patients to find appropriate biomarkers for this malignancy. However, due to the high mortality of this neoplasia, it is evident that more research on the search for possible biomarkers is necessary. What I consider that the following article should be published, making some improves.
I will mention my comments and suggestions below.
Introduction
Comment 1, line 36: I recommend that you add the Globocan reference for epidemiology
Comment 2, line 49: The authors mention the measurement of serum CA-125 for the diagnosis of ovarian cancer. However, currently, in the clinic, it is already done in conjunction with HE4, increasing its sensitivity.
Comment 3: The authors do not talk a bit about the background of miRNAs as biomarkers in cancer, I think they should add one or two lines about it.
Comment 4: They mention the objective of the study, which is to search for microRNAs in normal and neoplastic ovarian tissue. However, at the same time, it was done in serum samples, and that created confusion when reading the results. I think that they should mention this also in the objective of the study in the introduction section.
Results
Comment 1, Table 1: Within the characteristics of the patients, I consider that more data should be included such as the origin of the tumor, and menopausal status.
Comment 2, line 99: place the word Homo sapiens in italics
Comment 3, miRNAs in the serum of patients with OCC, are different from the patients who had ovarian tissue analyzed?
Comment 4, line 125: In the abstract, it is mentioned that 10 sets of specific primers were made, and, in the results, they mention that there were 12, so it does not agree.
Comment 5, figure 4: For better visualization of the results, put an asterisk to those that were statistically significant.
Discussion
Comment 1: The authors mention the findings of miRNAs found as biomarkers in other studies. However, it does not remain what was the new contribution to what was previously reported.
Comment 2: I consider that in this section perspectives of the study and its limitations should be included.
Author Response
We appreciate the time and effort provided by the editor and referees in reviewing our manuscript. We have addressed all issues indicated in the review report and hope that the revised version meets the journal's requirements for publication.
Response to Comments from Reviewer 3:
Comment 1:
The article identifies potential ovarian cancer-specific biomarkers for early diagnosis.
Numerous studies have evaluated expression profile microRNAs (miRNAs) in tissue and serum samples of ovarian cancer patients to find appropriate biomarkers for this malignancy. However, due to the high mortality of this neoplasia, it is evident that more research on the search for possible biomarkers is necessary. What I consider that the following article should be published, making some improves.
I will mention my comments and suggestions below.
Introduction
Comment 1: line 36: I recommend that you add the Globocan reference for epidemiology
Response: According to your suggestion, we added the data from GLOBOCAN 2020: “According to GLOBOCAN 2020, 314,000 women are diagnosed with ovarian cancer, and 207,000 women die from the same, every year. The incidence and mortality per 100,000 people are 6.6 and 4.2, respectively [1].” (page 1, line 39)
Comment 2: line 49: The authors mention the measurement of serum CA-125 for the diagnosis of ovarian cancer. However, currently, in the clinic, it is already done in conjunction with HE4, increasing its sensitivity.
Response: According to your suggestion, we added a sentence about the use of HE4 for ovarian cancer screening: “Recently, the use of human epididymis protein 4 (HE4) in conjunction with CA125 has achieved high sensitivity.” (page 2, line 55)
Comment 3: The authors do not talk a bit about the background of miRNAs as biomarkers in cancer, I think they should add one or two lines about it.
Response: According to your suggestion, we added these sentences about microRNA in the Introduction: “MiRNAs are 19 to 25 nucleotide-long endogenous molecules, which are integral to regulating gene expression. Aberrant miRNA expression profiles have been described in several cancers, and have been implicated as useful biomarkers which may aid cancer diagnostics and treatment [17].” (page 2, line 75)
Comment 4: They mention the objective of the study, which is to search for microRNAs in normal and neoplastic ovarian tissue. However, at the same time, it was done in serum samples, and that created confusion when reading the results. I think that they should mention this also in the objective of the study in the introduction section.
Response: According to your suggestion, we added this sentence explaining how we used blood samples: “Furthermore, serum samples obtained before and after surgery from patients with OCC and atypical endometrial hyperplasia (AEH) (controls) were compared to detect cancer-specific miRNAs in EVs.” (page 2, line 80)
Results
Comment 1: Table 1: Within the characteristics of the patients, I consider that more data should be included such as the origin of the tumor, and menopausal status.
Response: According to your suggestion, we added the origin of the tumor and menopausal status in the former Table 1, which has been renamed Table 3 and moved to the Materials and methods in response to a comment from another reviewer. (page 10, line 269)
Comment 2: line 99: place the word Homo sapiens in italics
Response: We placed the term Homo sapiens in italics as you suggested. (page 3, line 105)
Comment 3: miRNAs in the serum of patients with OCC, are different from the patients who had ovarian tissue analyzed?
Response: The patients from whom we obtained blood samples and OCC tissue are different. We wanted to use the blood samples from the same patients that we obtained OCC tissue from, however, RNA quality was not adequate for PCR. Our only option was to obtain the blood samples from other OCC patients.
Comment 4: line 125: In the abstract, it is mentioned that 10 sets of specific primers were made, and, in the results, they mention that there were 12, so it does not agree.
Response: We apologize for this typo. Among 17 detected miRNAs, we were able to create 12 specific primers. We revised the Abstract accordingly. (page 1, line 29)
Comment 5: figure 4: For better visualization of the results, put an asterisk to those that were statistically significant.
Response: According to your suggestion, we added asterisks to indicate statistical significance in Figure 4. (page 7)
Discussion
Comment 1: The authors mention the findings of miRNAs found as biomarkers in other studies. However, it does not remain what was the new contribution to what was previously reported.
Response: According to your suggestion, we added sentences about microRNA as biomarkers in the discussion section; “In the current study, the expression levels of miR-200c-3p, miR-200a-3p, miR200b-3p, miR-200b-5p, miR-200a-5p, miR-30a-5p, miR-30b-5p, and miR-21-5p in Te-EVs extracted from OCC tissues were significantly higher than those in normal ovaries. These eight miRNAs have not been previously identified as potential biomarkers for OCC.” (page 8, line 205)
Comment 2: I consider that in this section perspectives of the study and its limitations should be included.
Response: According to your suggestion, we added a paragraph on limitations to the discussion section:
“This study had several limitations. First, the control blood samples were obtained from AEH patients; AEH may have an influence on miRNAs. Second, the blood samples and OCC tissue specimens analyzed were obtained from two different groups of patients because of low RNA quality in blood samples obtained from the patients who provided tissue specimens. Third, the sample size was too small to prove that miRNA in EVs is a useful biomarker; it is not clear whether other differences between patients (age, infection, other tumor, etc.) would not result in changes in these miRNAs. Therefore, an analysis of a larger sample of OCC patients and age matched individuals without tumors is needed.” (page 9, line 235)

Reviewer 4 Report
In this manuscript the authors investigated the cancer-specific miRNAs as novel Ovarian clear cell carcinomas biomarkers, using tissue-exudative extracellular vesicles.
The manuscript can be accepted for publication if the authors are ready to incorporate the following revisions:
- In abstract, the authors reported that RNA sequencing was performed, but this is not described in the methods and not reported in the results, please correct.
- Figures 1, 2 and 5 must be mentioned after the figure, not before.
- The section 2.1 of Results should be moved to Materials and methods section.
Author Response
We appreciate the time and effort provided by the editor and referees in reviewing our manuscript. We have addressed all issues indicated in the review report and hope that the revised version meets the journal's requirements for publication.
Response to Comments from Reviewer 4:
Comment 1: In abstract, the authors reported that RNA sequencing was performed, but this is not described in the methods and not reported in the results, please correct.
Response: According to your suggestion, we revised this sentence in the Abstract: “Microarray analysis was performed to identify cancer-specific miRNAs in Te-EVs.” (page 1, line 24)
Comment 2: Figures 1, 2 and 5 must be mentioned after the figure, not before.
Response: According to your suggestion, the figures have been placed after their first mention in the manuscript.
Comment 3: The section 2.1 of Results should be moved to Materials and methods section.
Response: According to your suggestion, section 2.1 of Results was moved to the materials and methods section. (page 9, line 250)

Round 2
Reviewer 1 Report
The authors did not improve study design in any way. Without such analysis, it is impossible to evaluate the usefulness of these micro RNAs as biomarkers. Therefore, the work is too preliminary for publication. There needs to be at least some evaluation of variability in the general population before one can claim any use of these markers. Without this analysis, CA125 is much better to evaluate the presence of tumor in the patient.
Minor remark: The use of miR-16 as normalization marker for the qRT-PCR is not logical, as this marker is overexpressed in many tumors.
Author Response
We appreciate the time and effort provided by the editor and referees in reviewing our manuscript. We have addressed all issues indicated in the review report and hope that the revised version meets the journal's requirements for publication.
Response to Comments from Reviewer 1:
Comment 1: The authors did not improve study design in any way. Without such analysis, it is impossible to evaluate the usefulness of these micro RNAs as biomarkers. Therefore, the work is too preliminary for publication. There needs to be at least some evaluation of variability in the general population before one can claim any use of these markers. Without this analysis, CA125 is much better to evaluate the presence of tumor in the patient.
Response: The current study is preliminary one as you mentioned. Today, CA125 is useful marker to find ovarian cancers. However, we would like to find extremely- early staged ovarian cancer before elevation of CA125. Potentially, miRNA in EVs may enable this. As you mentioned, evaluation of variability in the general population must be needed for further study, however, we could not do them because of time- and cost-intensive. We intend to perform the same in our subsequent study.
Comment 2: Minor remark: The use of miR-16 as normalization marker for the qRT-PCR is not logical, as this marker is overexpressed in many tumors.
Response: According to your suggestion, we used RNU48 as the normalization control. The results were not different between using miR16 and RNU48. We revised the Figure 4 and sentences in method section. (page 7, line 150, figure 4; page 12, line 3059-360)